# Deep Generative Model for Periodic Graphs

**Shiyu Wang**
Emory University
shiyu.wang@emory.edu

**Xiaojie Guo**
IBM Thomas J. Watson Research Center
xguo7@gmu.edu

**Liang Zhao**[†]
Emory University
liang.zhao@emory.edu

## Abstract

Periodic graphs are graphs consisting of repetitive local structures, such as crystal nets and polygon mesh. Their generative modeling has great potential in real-world applications such as material design and graphics synthesis. Classical models either rely on domain-specific predefined generation principles (e.g., in crystal net design), or follow geometry-based prescribed rules. Recently, deep generative models have shown great promise in automatically generating general graphs. However, their advancement into periodic graphs has not been well explored due to several key challenges in 1) maintaining graph periodicity; 2) disentangling local and global patterns; and 3) efficiency in learning repetitive patterns. To address them, this paper proposes Periodical-Graph Disentangled Variational Auto-encoder (PGD-VAE), a new deep generative model for periodic graphs that can automatically learn, disentangle, and generate local and global graph patterns. Specifically, we develop a new periodic graph encoder consisting of global-pattern encoder and local-pattern encoder that ensures to disentangle the representation into global and local semantics. We then propose a new periodic graph decoder consisting of local structure decoder, neighborhood decoder, and global structure decoder, as well as the assembler of their outputs that guarantees periodicity. Moreover, we design a new model learning objective that helps ensure the invariance of local-semantic representations for the graphs with the same local structure. Comprehensive experimental evaluations have been conducted to demonstrate the effectiveness of the proposed method. The code and appendix is available in the Supplemental Material. The code of PGD-VAE is available at https://github.com/shi-yu-wang/PGD-VAE.

## 1 Introduction

Graphs is a ubiquitous data structure that describes objects and their relations. As a special type of graphs, periodic graphs are graphs that consist of repetitive local structures. They naturally characterize many real-world applications ranging from crystal nets containing repetitive unit cells to polygon mesh data containing repetitive grids (Figure 1) [51]. Hence, understanding, modeling, and generating periodic graphs have great potential in real-world applications such as material design and graphics synthesis. Different from general graphs, modeling generative process of periodic graphs requires to ensure the peri-

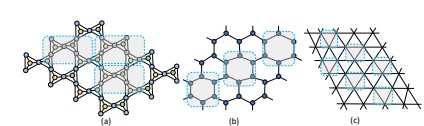

Figure 1: Examples of periodic graphs where basic units are highlighted: (a) structure of tridymite; (b) structure of graphene and (c) triangular grids [39]

---

† Corresponding author.

36th Conference on Neural Information Processing Systems (NeurIPS 2022).

odicity. Therefore, conventional graph models such as random graphs, small-world models cannot be used effectively. Classical studies on periodic graph generative models have a long history, where conventional methods can be categorized into two types: domain specific-based and geometry-based. Specifically, domain specific-based models generate periodic graphs relying on properties handcrafted based on domain knowledge [63, 5, 58]. In addition, geometry-based methods generate periodic graphs either according to predefined geometric rules to craft the targeted periodic graphs [22, 56] or simply gluing the initial fragment into periodic graphs [41].

As mentioned above, both domain specific-based and geometry-based generative models for periodic graphs require predefined properties that are either domain-specific or geometric-based, enabling them to fit well towards the properties that the predefined principles are tailored for, but usually they cannot do well for the others. For example, a geometry-based generative model can generate graphs within the predefined geometric rules but cannot generalize to other geometric properties. However, in many domains, network properties and generation principles are largely unknown, such as those explaining the thermal probability [53] and solubilities [19] of materials. The lack of predefined principles leads to the incapability of existing methods to model periodic graphs. Hence, it is imperative to fill the gap between the lack of powerful models and the complex generative process.

Expressive generative models that can directly learn the underlying data generative process automatically have been an extremely challenging task to accomplish, until very recent years when the advancement of deep generative models have exhibited great success in data such as images and graphs [49, 26, 30, 44, 18, 17]. Under this area, deep graph generative models aim at learning the distribution of general graph data representation extracted by graph neural networks [70]. However, despite the progress in graph neural networks for general graphs, they cannot be directly applied to handle periodic graphs due to the additional requirements of the generated graphs such as periodicity. It requires significant efforts to advance the current deep graph generative models in order to handle periodic graphs due to the following unique challenges: **(1) Difficulty in ensuring the periodicity of the generated graphs**. Existing works for generic graph generative models typically are not able to identify or generate the "basic unit", namely the repetitive local subgraphs. The conventional learning objective such as reconstruction errors or adversarial errors can only measure the generated graph quality globally but cannot ensure the identity of local structures as well as how different local structures are connected. **(2) How to jointly learn local and global topological patterns**. In periodic graph, there are two key patterns, namely the local structure which specifies the basic unit and the global topology which characterizes how local structures are connected to form the whole graph topology. Therefore, it is imperative for the generative models to be able to factorize these two different patterns for respective characterization and control. However, existing works for general graphs only learn a representation for the graph instead of disentangling them into different representations for local and global patterns; and **(3) Efficiency in learning large periodic graphs**. Due to the ubiquitousness of repetitive structures, periodic graphs are full of redundant information which makes existing works for generic graphs highly inefficient to model them. It is imperative to establish models that can automatically deduplicate the modeled generative process of periodic graphs.

To address these challenges, we developed Periodic-Graph Disentangled Variational Auto-encoder (PGD-VAE), a deep generative model for periodic graphs that can automatically learn, disentangle and generate local and global graph patterns. Specially, we proposed a novel encoder that contains a local-pattern encoder and a global-pattern encoder to disentangle the local and global semantics from the periodic graph representation. Besides, we designed a periodic graph decoder consisting of a local structure decoder, a global structure decoder, a neighborhood decoder and an assembler to assemble local and global patterns while preserving periodicity. To achieve the disentanglement of local and global semantics, we proposed a new learning objective that can simultaneously ensure the invariance of the local-semantic representations of graphs that have the same basic unit. Our contributions can be summarized as follows:

- We propose a new problem of end-to-end periodic graph generation by deep generative models.

- We develop PGD-VAE, a novel variational autoencoder that can decompose and assemble periodic graphs with the learned patterns of basic units, their pairwise relations, and their local neighborhoods.

- We propose a new learning objective to achieve the disentanglement of the local and global semantics by ensuring the invariance of the local-semantic representations of graphs that contain the same local structure.

- We conducted extensive experiments to demonstrate the effectiveness and efficiency of the proposed methods with multiple real-world and synthetic datasets.

## 2 Related works

### 2.1 Graph Representation Learning and Graph Generation

Surging development of deep-learning techniques, recent work on graph representation learning[33, 9, 34], especially graph neural networks (GNNs) [60, 74, 45, 31], is attracting considerable attention. Significant success has been achieved on developing various GNNs models to handle domain-specific graphs, such as traffic network [67, 69, 8], social network [72, 20, 15], knowledge graph [2, 12, 54] and materials structure [47, 61, 42]. Graph generation is one of the most important topics among the domain of graph representation learning. Due to the development of deep generative models, most GNN-based graph generation models borrow the framework of VAE [37, 38, 35, 59] or generative adversarial nets (GANs) [25, 68, 73]. For example, GraphVAE learns the distribution of given graphs and generates small graphs under the framework of VAE [52]. By contrast, MolGAN was proposed to generate small molecular graphs by means of GANs [13].

### 2.2 Disentangled Representation Learning

Disentangled representation learning has been attracting considerable attentions of the community, especially in the domain of computer vision [7, 55, 71, 32]. The goal of disentangled representation learning is to separate underlying semantic factors accounting for the variation of the data in the learned representation. This disentangled representation has been shown to be resilient to the complex factors involved [3, 16], and can enhance the generalizability and improve robustness against adversarial attack [27, 1]. Intuitively, disentangled representation learning can achieve superior interpretability over regular graph representation learning tasks to better understand the graphs in various domains [28]. This motivates the surge of a few VAE-based approaches that modify the VAE objective by adding, removing or adjusting the weight of individual terms for deep graph generation tasks [27, 10, 36]. Moreover, Martinkus et al. [46] proposed GAN-based model global and local graph structure via graph spectrum. Disentangled representation learning is important in modeling periodical graphs where the process requires to distinguish the repeatable patterns from the others.

### 2.3 Periodical Graph Learning

Periodic graph is the graph that consists of repetitive basic units in its structure, and can fit into the data structure in a number of domains, such as crystal structures in materials science [4, 14, 62, 24], polygon mesh in computer graphics [21] and motion planning in robotics [23]. Several graph representation learning tasks have been conducted on this type of graph. For example, periodic graph-structured information in the airport network can be formatted into a periodic graph and a graph convolutional network-based (GCN-based) was developed for flight delay prediction [6]. The multiple periodic spatial-temporal graphs of human mobility was employed to learn the community structure based on a collective embedding model [57]. In addition, graph generation model in periodic graphs can be categorized in to two scenarios: (1) domain specific-based models and (2) geometry-based models. For example, CDVAE borrows specific bonding preferences between different atoms and generates the periodic structure of materials in a diffusion process [63]. Moreover, [48] achieved geometric construction of periodic graphs by projecting two-dimensional hyperbolic tilings onto triply periodic minimal surfaces. They both need predefined rules to generate valid periodic graphs. Besides, the periodic graphs are usually large, the large time or space complexity of existing graph generative models limits the advance of periodic graph generation, motivating us to propose an end-to-end periodic graph generative model, PGD-VAE, that bears superior scalability on large periodic graphs.

## 3 Methodology

In this section, the problem formulation is first provided before moving on to derive the overall objective, following which the novel encoder of PGD-VAE that can respectively extract local and global information of the graph structure is introduced. Afterwards, the decoder that generates local and global structure, and the algorithm of the assembler are described.

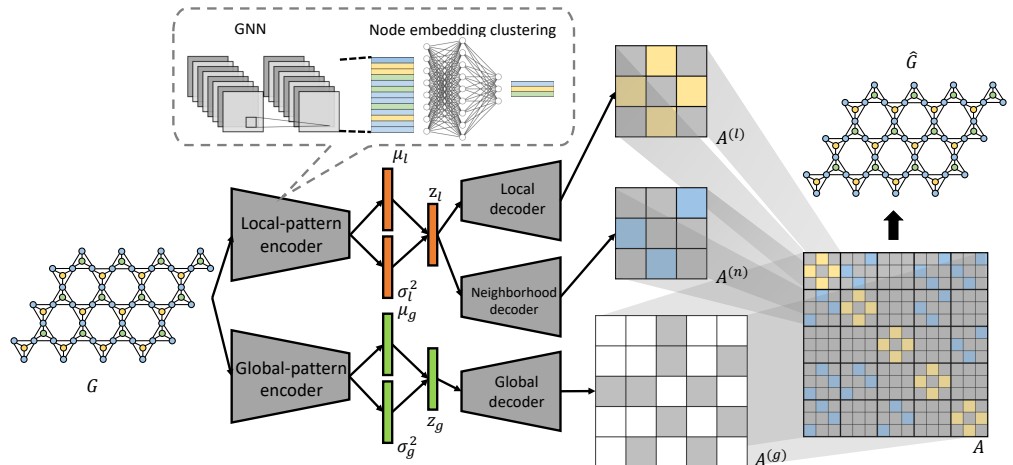

Figure 2: Overview of PGD-VAE. The graph representation $z_l$ and $z_g$ that respectively capture local and global semantics are learned by local-pattern and global-pattern encoders, respectively. Then $A^{(l)}$, $A^{(n)}$ and $A^{(g)}$ are generated by local decoder, neighborhood decoder and global decoder, respectively. Finally, the generated graph is obtained by assembling $A^{(l)}$, $A^{(n)}$ and $A^{(g)}$. Note that $A$ sticks to the intrinsic node ordering and permutation naturally followed by the graphs in the training set.

## 3.1 Notations and Problem Definition

### 3.1.1 Periodic graphs

Define a periodic graph as $G = (\mathcal{V}, \mathcal{E}, A)$, where $\mathcal{V}$ is the set of nodes and $\mathcal{E} \subseteq \mathcal{V} \times \mathcal{V}$ is the set of edges of the graph $G$. The repeated local structures can be defined as basic units of size $n$ (i.e., number of nodes is $n$). Suppose the periodic graph $G$ consists of $m$ basic units as well as their connections, this means the node set $\mathcal{V} = \cup_{u=1}^{m} v_{i,u}$ is the union of nodes in all basic units, and $\mathcal{E} = \{\cup_{u=1}^{m} e_{ij,u}\} \cup \{\cup_{u \neq v}^{m} e_{ij,i \in u, j \in v}\}$ is the union of all edges within and across basic units of the graph. $A \in \mathbb{R}^{N \times N}$ corresponds to the adjacency matrix of the graph, in which $A_{ij} = 1$ if $e_{ij} \in \mathcal{E}$ and $A_{ij} = 0$ otherwise. In addition, we define three matrices: (1) $A^{(l)} \in \mathbb{R}^{n \times n}$ is the adjacency matrix of the basic unit of the graph; (2) $A^{(g)} \in \mathbb{R}^{m \times m}$ is the adjacency matrix denoting the pairwise relations among basic units; and (3) $A^{(n)} \in \mathbb{R}^{n \times n}$ is the incidence matrix regarding how the nodes in adjacent basic units are connected. Without loss of generality, all the adjacency matrices stick to a specific and intrinsic node ordering (i.e., permutation) where $A^{(l)}$'s occupy the diagonal blocks in $A$. The reason that we say it is "intrinsic" is because such node ordering is usually naturally provided by the data (see the dataset description in Section 4.1 for details).

### 3.1.2 Represent $A$ by $A^{(l)}$, $A^{(g)}$, and $A^{(n)}$

As the adjacency matrix $A$ of the periodic graph contains repeated local structures, we want to use an alternative "deduplicated" way to represent $A$ efficiently. This will be achieved by concisely representing $A$ with a function of $A^{(l)}$, $A^{(g)}$, and $A^{(n)}$: $A = f(A^{(l)}, A^{(g)}, A^{(n)})$, which is detailed in the following and illustrated in the right part of Figure 2. First, define a binary block matrix $P \in \{0, 1\}^{nm \times m}$ s.t. $P \cdot P^T = J$, where $J$ is the block diagonal matrix whose diagonal blocks are all 1's. Then, $\hat{A}^{(g)}$, the augmentation of $A^{(g)}$ can be achieved as:

$$\hat{A}^{(g)} = PA^{(g)}P^T. \tag{1}$$

Additionally, we define the matrix $Q \in \mathbb{R}^{nm \times n}$ as the concatenation of $m$ identity matrices of size $n$. Then we can obtain $\hat{A}^{(l)}$ and $\hat{A}^{(n)}$, the repetition of $A^{(l)}$ and $A^{(n)}$, respectively, along both dimensions by $m$ :

$$\hat{A}^{(l)} = QA^{(l)}Q^T, \hat{A}^{(n)} = QA^{(n)}Q^T. \tag{2}$$

Consequently, we can deterministically obtain $A$ from $A^{(l)}$, $A^{(g)}$ and $A^{(n)}$ via the function $f$:

$$A = f(A^{(l)}, A^{(g)}, A^{(n)}) = (M^{(n)} \odot QA^{(n)}Q^T + (M^{(n)} \odot QA^{(n)}Q^T)^T) \odot PA^{(g)}P^T + M^{(l)} \odot QA^{(l)}Q^T$$
$$= (M^{(n)} \odot \hat{A}^{(n)} + (M^{(n)} \odot \hat{A}^{(n)})^T) \odot \hat{A}^{(g)} + M^{(l)} \odot \hat{A}^{(l)}, \qquad (3)$$

where $M^{(l)}$ and $M^{(n)}$ are two mask matrices. Specifically, the off-diagonal blocks of size $n$ in $M^{(l)}$ are 0's while the diagonal blocks in $M^{(l)}$ are 1's. We also specify that the upper-triangular blocks of size $n$ in $M^{(n)}$ are 1's and other blocks in $M^{(n)}$ are 0's. In the following, we may exchangeably use $G$ and $A$ to denote the whole graph since it is determined by its topology.

### 3.1.3 Deep generative models of periodic graphs

Learning generative models for a periodic graph aims to learn the conditional distribution $p(G|Z)$. Define $Z = (z_l, z_g)$ as the learned graph representation based on the structure of graphs. $z_l$ is designed to capture the local semantic factors that characterize the topological structure of basic units, while $z_g$ is designed to represent the global semantics on how basic units are topologically connected.

## 3.2 Learning Deep Periodic Graph Generative models

We first introduce the variational inference of the generative model and learning objective. Then we introduce our local and global encoders, as well as the new periodic graph decoder.

### 3.2.1 Periodic graph generative model inference

As discussed in the problem definition, we aim to learn the conditional distribution of $G$ given $Z = (z_l, z_g)$. So we aim to maximize the marginal likelihood of the observed periodic graph $G$ in expectation over the distribution of the latent variables $(z_l, z_g)$. For a given periodic graph $G$, we define the prior distribution of the latent representation as $p(z_l, z_g)$ which is intractable to infer. As a result, we leverage variational inference, in which the posterior distribution is approximated by $q_\phi(z_l, z_g|G)$. Hence, another goal is to minimize the Kullback–Leibler (KL) divergence between the true prior and the approximated posterior distributions. To encourage the disentanglement of $z_l$ and $z_g$, we introduce a constraint by matching the inferred posterior configurations of the latent factors $q_\phi(z_l, z_g|G)$ to the prior distribution $p_\theta(z_l, z_g)$:

$$\max_{\theta, \phi} \mathbb{E}_{G \sim D}[\mathbb{E}_{q_\phi(z_l, z_g|G)} \log p_\theta(G|z_l, z_g)]$$

$$s.t. \mathbb{E}_{G \sim D}[KL(q_\phi(z_l, z_g|G)||p_\theta(z_l, z_g))] < I, \qquad (4)$$

where $D$ refers to the observed graphs and $I$ is a constant. The detailed derivation of the VAE objective can be seen at Appendix **??**. Then, we can decompose the constraint term in Eq. 4 given the assumption that $z_l$ and $z_g$ are independent, leading to:

$$p_\theta(z_l, z_g) = p_\theta(z_l)p_\theta(z_g), q_\phi(z_l, z_g|G) = q_\phi(z_l|G)q_\phi(z_g|G) \qquad (5)$$

Based on the problem formulation, $G$ can be represented as $\{A^{(l)}, A^{(g)}, A^{(n)}\}$. Since $z_l$ should capture and only capture the local patterns, thus $A^{(l)}$ and $A^{(n)}$ are dependent on $z_l$ and conditionally independent from each other given $z_l$, namely $A^{(l)} \perp A^{(n)}|z_l$. And $z_g$ should capture and only capture the global patterns, we have

$$p_\theta(G|z_l, z_g) = p_\theta(A^{(l)}, A^{(n)}|z_l)p_\theta(A^{(g)}|z_g) = p_\theta(A^{(l)}|z_l)p_\theta(A^{(n)}|z_l)p_\theta(A^{(g)}|z_g). \qquad (6)$$

Then, the objective function can be written as:

$$\max_{\theta, \phi} \mathbb{E}_{G \sim D}[\mathbb{E}_{q_\phi(z_l, z_g|G)} \log p_\theta(A^{(l)}|z_l)p_\theta(A^{(n)}|z_l)p_\theta(A^{(g)}|z_g)]$$

$$s.t. \mathbb{E}_{G \sim D}[KL(q_\phi(z_l|G)||p_\theta(z_l))] < I_l; \mathbb{E}_{G \sim D}[KL(q_\phi(z_g|G)||p_\theta(z_g))] < I_g, \qquad (7)$$

where $I_l$ and $I_g$ are the information bottleneck to control the information captured by the latent vector $z_l$ and $z_g$. Based on the objective we derived above, we propose a novel objective for PGD-VAE by extending Eq. 7 in order to not only fulfill the optimization objective of the general VAE framework (Eq. 4), but also secure the disentanglement of local and global semantics of the graph. Specifically, we force the $z_l$ equal for graphs with the same basic unit (Eq. 8). At the meanwhile, we distinguish $z_l$'s of graphs with different basic units (Eq. 8).

$$\max_{\theta, \phi} \mathbb{E}_{G \sim D}[\mathbb{E}_{q_\phi(z_l, z_g|G)} \log p_\theta(A^{(l)}|z_l)p_\theta(A^{(n)}|z_l)p_\theta(A^{(g)}|z_g)]$$

$$s.t. \mathbb{E}_{G \sim D}[KL(q_\phi(z_l|G)||p_\theta(z_l))] < I_l; \mathbb{E}_{G \sim D}[KL(q_\phi(z_g|G)||p_\theta(z_g))] < I_g \qquad (8)$$

$$z_{l,i}^{(j)} = z_{l,i}, j = 1, ..., N_i, i = 1, ..., N; z_{l,s} \neq z_{l,t}, s = 1, ..., N, t = 1, ..., N, s \neq t$$

where $z_{l,i}^{(j)}$ and $z_{g,i}^{(j)}$ are respectively the local and global representations of the $j$-th graph for the $i$-th basic unit. The derived objective function for PGD-VAE (Eq. 8) can meet the objective in our problem formulation, as stated in Theorem 3.1. The proof of Theorem 3.1 is provided in Appendix **??**.

**Theorem 3.1.** *The proposed objective function in Eq. 8 guarantees that $z_l$ and $z_g$ satisfy:*

1. *$z_l$ capture and only capture the local semantic of the graph;*

2. *$z_g$ capture and only capture the global semantic of the graph.*

### 3.2.2   Learning objectives

To optimize the overall objective in Eq. 8 while satisfying the constraint, we transform the inequality constraint into tractable form. Given that $I_l$ and $I_g$ are constants, we rewrite the first two constraints of Eq. 8 using Lagrangian algorithm under KKT condition as:

$$L_{KL} = -\beta_1 KL\{q_\phi(z_l|\mathbf{G})||p_\theta(z_l)\} - \beta_2 KL\{q_\phi(z_g|\mathbf{G})||p_\theta(z_g)\}, \tag{9}$$

where $\mathbf{G}$ represents a set of observed graphs, the Lagrangian multipliers $\beta_1$ and $\beta_2$ are regularization coefficients that constraint the capacity of latent information channels $z_l$ and $z_g$, respectively, and put implicit pressure of independence on the learned posterior.

Additionally, the goal of the last two constraints of Eq. 8 is to enforce the graphs that contain the same repeat patterns (i.e., local information) sharing the same $z_l$ and the graphs that contain different repeat patterns having different $z_l$. Thus, to both minimize the distances of $z_l$ among the graphs with the same unit cell and maximize the distances of $z_l$ among the graphs with different unit cells, we propose to minimize a contrastive loss $L_{contra}$ as:

$$L_{contra} = -\beta_3 \sum_{i=1}^{N} \sum_{j=1}^{N_i} \sum_{k=1}^{N_i} \log \frac{exp\{sim(z_{l,i}^{(j)}, z_{l,i}^{(k)})/\tau\}}{\sum_{s=1,s\neq i}^{N} \sum_{t=1}^{N_s} exp\{sim(z_{l,i}^{(j)}, z_{l,s}^{(t)})/\tau\}},$$

where $\tau$ is a predefined temperature parameter, and $\beta_3$ is the weights added on the contrastive loss to adjust the convergence of $z_l$ in graphs with the same basic unit and the divergence in graphs with different basic units. $sim()$ is the cosine similarity borrowed in computing the contrastive loss [66].

In summary, the final implemented loss function minimized $L$ includes three parts: (1) $L_{rec}$ to minimize the reconstruction loss of the input and generated graphs; (2) $L_{KL}$ to minimize the KL divergence of the posterior and its approximation; and (3) $L_{contra}$, the contrastive loss to force $z_l$ equal for graphs with the same basic unit and different for those with different basic units:

$$L = -\mathbb{E}_{G\sim D}[\mathbb{E}_{q_\phi(z_l, z_g|G)} \log p_\theta(A^{(l)}|z_l)p_\theta(A^{(n)}|z_l)p_\theta(A^{(g)}|z_g)] + L_{KL} + L_{contra} = L_{rec} + L_{KL} + L_{contra}$$

### 3.3   Architecture of PGD-VAE

Based on the above periodic graph generative model and its learning objective, we are proposing our new Periodic Graph Disentangled VAE model (PGD-VAE). In general, PGD-VAE takes the whole adjacency matrix $A$ as the input and can automatically identify $A^{(l)}$, $A^{(g)}$ and $A^{(n)}$ by model architecture and learning objective. Specifically, PGD-VAE contains a local-pattern encoder and a global-pattern encoder to disentangle the graph representation into local and global semantics, respectively. It also contains a local structure decoder, a global structure decoder, a neighborhood decoder together with an assembler to formalize the graph while ensuring the periodicity.

An overview of PGD-VAE is shown in Figure 2. The proposed framework has two encoders, each of which models one of the distributions $q_\theta(z_g|G)$ and $q_\theta(z_l|G)$; and three decoders to model $p_\theta(A^{(l)}|z_l)$, $p_\theta(A^{(n)}|z_l)$, and $p_\theta(A^{(g)}|z_g)$, respectively. Each type of representations is sampled using its own inferred mean and standard derivation. For example, the representation vectors $z_l$ are sampled as $z_l = \mu_l + \sigma_l * \eta$, where $\eta$ follows a standard normal distribution.

### 3.3.1   Encoder

The encoder of PGD-VAE is designed to learn $z_l$ and $z_g$ such that $z_l$ captures only the local semantics of the graph and $z_g$ captures only the global semantics of the graph. Thus, we propose a local-pattern encoder to encode the local information and a global-pattern encoder to encode the global information, as illustrated in Figure 2.

**Local-pattern encoder** We designed the encoder to learn an assignment matrix $A_{rep} \in \mathcal{R}^{C \times N}$ to identify representative nodes of the graph (Eq. 10) where $C$ is predefined as the number of

representative nodes to be assigned. This node embedding clustering module can enforce $z_l$ to only capture the local information, namely excluding the global information (e.g., size of the graph) from $z_l$, because: 1) Each node embedding preserves local pattern since GIN embeds local information of nodes; 2) Since node embedding of periodic graph is periodic, embedding of a node in a basic unit is the same as the corresponding node in another basic unit. By clustering all node embedding, each cluster preserves repetitive node embedding patterns across different local regions. In our setting, these representative nodes capture different embedding with each other. The column of $A_{rep}$ corresponding to the node $v$ can be calculated:

$$A_{rep,v} = SOFTMAX(MLP(h_v^{(K)}|v \in \mathcal{V})) \tag{10}$$

where $h_v^{(K)}$ is the embedding of node $v$ at the last layer of Graph Isomorphism Network (GIN) [64]. GIN is provably the most expressive among the class of GNNs and is as powerful as the WeisfeilerLehman graph isomorphism test [64]. To preserve all information contained by the representative nodes, We concatenate all representative node embedding to obtain $\mu_l$ and $\sigma_l$:

$$\mu_l = MLP(CONCATE(D_{A_{rep}}^{-1} \cdot A_{rep} \cdot h_v^{(K)}|v \in \mathcal{V}))$$
$$\sigma_l = MLP(CONCATE(D_{A_{rep}}^{-1} \cdot A_{rep} \cdot h_v^{(K)}|v \in \mathcal{V})), \tag{11}$$

where $D_{A_{rep}}$ has the degree of each cluster in its diagonal. $z_l$ can then be sampled according the VAE framework: $z_l \sim \mathcal{N}(\mu_l, \sigma_l)$.

**Global-pattern encoder** To enforce that $z_g$ captures all the global information, GIN is leveraged to generate node-level embedding of the graph. Additionally, PGD-VAE leverages sum aggregator as the graph-level readout (Eq. 12) to obtain the graph-level embedding $z_g$ while achieving maximal discriminative power [64].

$$\mu_g = SUM(MLP(h_v^{(K)}|v \in \mathcal{V})), \sigma_g = SUM(MLP(h_v^{(K)}|v \in \mathcal{V})) \tag{12}$$

The $h_v^{(K)}$ represents the embedding of node $v$ obtained at the last layer of GIN. $z_g$ can then be sample according to the VAE framework: $z_g \sim \mathcal{N}(\mu_g, \sigma_g)$.

### 3.3.2 Decoder

As shown in Figure 2, the decoder contains four parts: (1) the local-structure decoder to generate $A^{(l)}$; (2) the global-structure decoder to generate $A^{(g)}$; (3) the neighborhood decoder to generate $A^{(n)}$; and (4) an assembler $f : \{A^{(l)}, A^{(g)}, A^{(n)}\} \rightarrow A$ which generates $A$ according to Eq. (3). The local structure decoder takes the local-semantic representation $z_l$ as the input to generate the structure of the basic unit. The global structure decoder takes the global-semantic representation $z_g$ as the input to generate the global organization of basic units. Since how basic units are connected is regarded as the local structure of the graph, the neighborhood decoder takes the local-semantic representation $z_l$ as the input to generate the connection between adjacent basic units. The assembler $f$ is a closed-form function defined in Section 3.1 to assemble $A^{(l)}$, $A^{(g)}$ and $A^{(n)}$ into $A$:

$$A = f(A^{(l)}, A^{(g)}, A^{(n)}). \tag{13}$$

Here $f_l$, $f_g$, $f_n$ are deterministic functions serving as three decoders to generate $A^{(l)}$, $A^{(g)}$ and $A^{(n)}$, respectively. They are approximated by three multilayer perceptrons (MLPs) in implementation. Edges are sampled from the Bernoulli distribution given the probability output of MLP.

### 3.4 Complexity Analysis

We theoretically evaluate the scalability of our model and comparison models in the number of generation steps, similar to the analysis in [43]. Since the decoder of PGD-VAE follows an one-shot strategy using MLP, it requires $O(1)$ generation steps. Additionally, the space complexity of PGD-VAE is $O(max\{n^2, m^2\})$, since the generated graph is determined by basic units, the connection of adjacent basic units and the global pattern. The graph size can be calculated as $mn$. The number of generation steps of PGD-VAE is aligned with that of VGAE, but is lower than that of GraphVAE and GRAN. Nevertheless, PGD-VAE has the smallest space complexity. For time complexity, PGD-VAE can achieve $O(m^2+n^2+mn)$ by using matrix replication and concatenation tricks in implementation, better than that of other comparison models. The number of generation steps, time complexity and the space complexity of PGD-VAE and comparison models are summarized in Table 1.

Table 1: Complexity of PGD-VAE and comparison models ($n$ is the size of the basic unit, $m$ is the size of the global pattern)

| Model | Number of generation steps | Time complexity | Space complexity |
|---|---|---|---|
| VGAE | $O(1)$ | $O((mn)^2)$ | $O((nm)^2)$ |
| GraphRNN | $O((nm)^2)$ | $O((mn)^2)$ | $O((nm)^2)$ |
| GRAN | $O(nm)$ | $O((mn)^2)$ | $O((nm)^2)$ |
| PGD-VAE | $O(1)$ | $O(m^2 + n^2 + mn)$ | $O(max\{n^2, m^2\})$ |

Table 2: PGD-VAE compared to state-of-the-art deep generative models on QMOF, MeshSeg and Synthetic datasets using KLD (KLD_cluster: KLD of clustering coefficient, KLD_dense: KLD of graph density, PGD-VAE-1: PGD-VAE without regularization, PGD-VAE-2: PGD-VAE replacing GIN with GCN in encoder, PGD-VAE-3: PGD-VAE with single MLP as decoder)

| Method | QMOF | | MeshSeq | | Synthetic | |
|---|---|---|---|---|---|---|
| | KLD_cluster | KLD_dense | KLD_cluster | KLD_dense | KLD_cluster | KLD_dense |
| VGAE[38] | 1.7267 | 1.5168 | 0.9723 | 0.9642 | 4.6196 | 1.4222 |
| GraphVAE[52] | 2.2093 | 2.8441 | 1.1365 | 1.1236 | 3.5928 | 0.9051 |
| GraphRNN[65] | 3.1055 | 5.0048 | 1.4316 | 1.0271 | 6.4719 | 3.6064 |
| GRAN[43] | 3.3107 | 2.7208 | 1.2466 | 0.8879 | 5.7501 | 2.3020 |
| PGD-VAE-1 | 1.0193 | 1.0250 | 0.4630 | 0.4921 | 3.2672 | 0.8399 |
| PGD-VAE-2 | **0.8508** | **0.9332** | 0.4539 | 0.4862 | 4.1596 | 2.7684 |
| PGD-VAE-3 | 0.9597 | 1.0224 | 0.5496 | 0.5099 | 3.3850 | 0.9366 |
| PGD-VAE | 1.0164 | 1.0512 | **0.3977** | **0.4535** | **3.1863** | **0.6824** |

Table 3: PGD-VAE compared to state-of-the-art deep generative models on QMOF, MeshSeg and Synthetic datasets according to uniqueness and novelty (PGD-VAE-1: PGD-VAE without regularization, PGD-VAE-2: PGD-VAE replacing GIN with GCN in encoder, PGD-VAE-3: PGD-VAE with single MLP as decoder)

| Method | QMOF | | MeshSeq | | Synthetic | |
|---|---|---|---|---|---|---|
| | Uniqueness | Novelty | Uniqueness | Novelty | Uniqueness | Novelty |
| VGAE[38] | **100%** | **100%** | **100%** | **100%** | **100%** | **100%** |
| GraphVAE[52] | **100%** | **100%** | **100%** | **100%** | **100%** | **100%** |
| GraphRNN[65] | 23.34% | **100%** | 71.39% | **100%** | 26.76% | **100%** |
| GRAN[43] | 8.00% | **100%** | 15.33% | **100%** | 18.67% | **100%** |
| PGD-VAE-1 | **100%** | **100%** | **100%** | **100%** | **100%** | **100%** |
| PGD-VAE-2 | **100%** | **100%** | **100%** | **100%** | **100%** | **100%** |
| PGD-VAE-3 | **100%** | **100%** | **100%** | **100%** | **100%** | **100%** |
| PGD-VAE | **100%** | **100%** | **100%** | **100%** | **100%** | **100%** |

## 4 Experiments

This section reports the results of both quantitative and qualitative experiments that were performed to evaluate PGD-VAE with other comparison models. Two real-world datasets and one synthetic datasets of periodic graphs were employed in the experiments. All experiments were conducted on the 64-bit machine with an NVIDIA GPU, NVIDIA GeForce RTX 3090.

### 4.1 Data

Two real-world datasets and one synthetic dataset were employed to evaluate the performance of PGD-VAE and other comparison models. All the comparison and our proposed methods only take the whole adjacency matrix $A$. The information of $A^{(l)}$, $A^{(g)}$, and $A^{(n)}$ are hidden from all the methods. Also, in all the datasets, the adjacency matrix of each graph followed its intrinsic permutation when they were generated, namely all the diagonal blocks are occupied by the basic unit $A^{(l)}$, as shown in Figure 2. So all the graphs generated by all the models trained by these datasets also follow such fixed permutation which naturally saves them from considering node permutation. 1) **QMOF** is a publicaly available database of computed quantum-chemical properties and molecular structures [50]. We selected metal–organic frameworks (MOFs) from QMOF consisting of unit cells as the basic unit (i.e., $A^{(l)}$). $A^{(g)}$ and $A^{(n)}$ can be automatically determined by Atomic Simulation Environment (ASE) [40]. Finally, the whole adjacency matrix $A$ was the assemble of $A^{(l)}$, $A^{(g)}$ and $A^{(n)}$ via Eq. 3. 2) **MeshSeg** contains 380 meshes for quantitative analysis of how people decompose objects into parts and for comparison of mesh segmentation algorithms [11]. We extracted meshes from MeshSeq and made 10 replicates for each mesh. The whole adjacency matrix $A$ was directly obtained from the graphs in this dataset. 3) **Synthetic dataset**. Three synthetic datasets were generated based on three different basic units (i.e., $A^{(l)}$), namely triangle, grid and hexagon. For each of the above basic unit types, different numbers of basic units were used to compose the whole graph (i.e., represented by

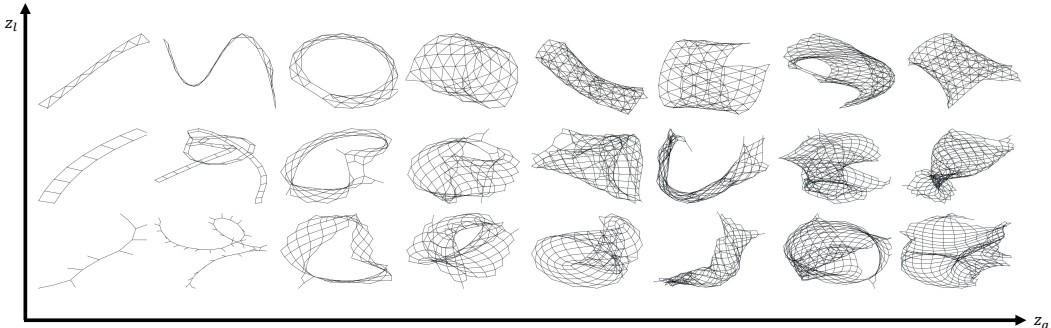

Figure 3: Visualizing variations of generated periodical graphs regarding local and global semantics

the whole adjacency matrix $A$) via Equation (4), with self-designed $A^{(g)}$ and $A^{(n)}$. All adjacency matrices were padded into a fixed size with zeroes in implementation. The statistics of the three datasets have been summarized in Appendix, Table **??** and more details can be found in Appendix **??**.

## 4.2    Comparison Models

Four state-of-the-art deep generative models were employed as comparison models to compare with PGD-VAE[1]: (1) *VGAE*, a VAE-based graph generative model [38] by learning the graph-level representation and using MLP to generate the adjacent matrix; (2) *GraphVAE*, a VAE-based graph generative model [52] by modeling the existence of nodes and edges, as well as their attributes as independent random variables in a probabilistic graph;(3) *GraphRNN*, a deep autoregressive model that accounts for non-local dependencies of edges by decomposing generation process into a sequence of node and edge conformations [65]; and (4) *GRAN*, a deep autoregressive model [43] which generates a block of nodes and associated edges at a time, and captures the auto-regressive conditioning between the already-generated and to-be-generated pieces of the graph using GNNs with attention. PGD-VAE and all comparison methods only take the whole adjacency matrix $A$ as input, without extra information on $A^{(l)}$, $A^{(g)}$, or $A^{(n)}$. Noticeably, we used fixed permutation that restricts the adjacency matrix $A$ to fixed node ordering as illustrated in Figure 2. Graphs with predefined permutation can easily be provided in periodic graph datasets as aforementioned.

In addition, three ablation study are conducted by evaluating four variants of PGD-VAE: (1) PGD-VAE without regularization term; (2) PGD-VAE replacing GIN with GCN in the encoder; (3) PGD-VAE with a single MLP as decoder. The comparison models are implemented using their default settings. The implementation details of PGD-VAE are presented in Appendix **??**.

## 4.3    Evaluating the Generated Graphs

**Quantitative evaluation**. To quantitatively evaluate generation performance of PGD-VAE and comparison models, we compute the KL Divergence (KLD) between generated and ground-truth graphs to measure the similarity of their distributions regarding: (1) average clustering coefficient and (2) density of graphs [29]. Moreover, we adopt uniqueness and novelty to evaluate generated graphs. The uniqueness measures the diversity of generated graphs. The novelty reflects whether the model has generated unseen graphs by measuring the proportion of generated graphs that are not in the set of training graphs. The KLD-based results were calculated and shown in Table 2. The results show that, PGD-VAE achieves KLD smaller than comparison models by 0.8191 for clustering coefficient and 0.5062 for graph density on average on MeshSeq dataset, and 2.4275 for clustering coefficient as well as 1.7611 for graph density on average on Synthetic dataset. Interestingly, PGD-VAE without disentanglement (PGD-VAE-1) achieves 3.2672 and 0.8399 of KLD on clustering coefficient and density for Synthetic dataset, which is worse than the full model by 0.0809 and 0.1575 as well. In addition, on QMOF dataset, PGD-VAE that replaces GIN with GCN (PGD-VAE-2) achieves KLD of 0.8508 regarding clustering coefficient and 0.9332 regarding graph density, outperforming

---

[1]Some graph generative models, such as GraphVAE that employs $O(n^4)$ on its max-pooling matching algorithm, are not included in the comparison due to its poor scalability to large graphs, which is common in QMOF and MeshSeg datasets.

other models, which is likely due to the fact that the basic unit of MOF is usually more complex and needs more powerful GNN to capture the local information of the graph. Also, the PGD-VAE (PGD-VAE-3) with a single MLP as the decoder hurts the performance of the model (Table 2), indicating the efficiency of the proposed hierarchical decoder. The measuring of uniqueness and novelty of generated graphs by different models has been summarized in Table 3. PGD-VAE and all its variants achieve better performance than GraphRNN and GRAN, as they tend to generate unique graphs, which is desired for tasks such as materials designing and object synthesis.

**Qualitative evaluation for disentanglement** To validate the disentanglement of local and global semantics in the graph representation, we qualitatively explore whether PGD-VAE can uncover and disentangle these two latent factors. To this end, we continuously changing the value of $z_l$ or $z_g$ while keeping another latent vector fixed and simultaneously visualizing the change of generated graphs. Figure 3 shows the variation of generated graphs when traversing two latent factors $z_l$ and $z_g$.

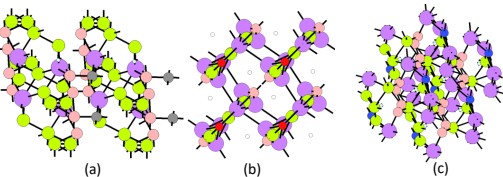

Figure 4: Periodic graphs of MOFs generated by PGD-VAE. Nodes of the graph are colored by their degrees representing atoms of the MOF. Edges represent chemical bonds of the MOF: (a) structure of $C16H8Na8Cu4N16S16$ composed of $C$, $H$, $Na$, $Cu$, $N$ and $S$ atoms; (b) structure of $C24H32Cl8Cu4N8$ which has a different basic unit consisting of $C$, $H$, $Cl$, $Cu$ and $N$ atoms; (c) structure of $C16H16Mg8O20S20$ composed of a relatively large basic unit with $C$, $H$, $Mg$, $O$ and $S$ atoms.

**A case study of generating MOFs** We showcase several representative periodic graph structures generated by the proposed PGD-VAE for MOF design in this section. As shown in Figure 4, there are three diverse and novel generated MOFs. The first one can be interpreted as $C16H8Na8Cu4N16S16$, for which PGD-VAE have generated the basic unit (i.e., $C4H2Na2CuN4S4$) that are connected with each other by incident chemical bonds. The second one can be interpreted as $C24H32Cl8Cu4N8$, which is highly different from the first one in terms of atom diversity. The third generated MOF is $C16H16Mg8O20S20$, which has a larger basic unit, namely, $C4H4Mg2O5S5$. In addition, the basic units of these generated structures are all constructed in a periodic way with high validity and diversity, which demonstrates the effectiveness of the proposed PGD-VAE. More details of the MOF decoration and visualization are provided in Appendix **??**.

**Evaluating scalability** We evaluated the time complexity of PGD-VAE and comparison models by running 10 epochs on graphs in stratified synthesis dataset. Expectedly, all comparison models consume more time than PGD-VAE. The details are presented in Appendix **??**.

## 5 Conclusion

We propose PGD-VAE, a novel deep generative model for periodic graphs that can automatically learn, disentangle, and generate local and global patterns. Our method can guarantee the periodicity of generated graphs and outperforms existing graph generative models in periodic graph generation, both quantitatively and qualitatively. In the future, we attempt to introduce the node and edge features into the model to advance domains such as materials design and object synthesis.

## Acknowledgments and Disclosure of Funding

This work was supported by the NSF Grant No. 2007716, No. 2007976, No. 1942594, No. 1907805, No. 1841520, No. 1755850, Meta Research Award, NEC Lab, Amazon Research Award, NVIDIA GPU Grant, and Design Knowledge Company (subcontract number: 10827.002.120.04).

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
