# A Derivation of variational inference

The posterior distribution of the latent variable is $p(z|G) = \frac{p(G|z)p(z)}{p(G)}$, where $p(G)$ is intractable. Hence we approximate it via $q_\phi(z|G)$ by minimizing $KL(q_\phi(z|G)||p(z|G)) = -\int q_\phi(z|G) \log \frac{p(G|z)p(z)}{q_\phi(z|G)} dz + \log p(G)$. Since $G$ is given, then minimizing $KL(q_\phi(z|G)||p(z|G))$ is equivalent to maximizing the evidence lower bound (ELBO): $\int q_\phi(z|G) \log \frac{p(G|z)p(z)}{q_\phi(z|G)} dz = \mathbb{E}_{q_\phi(z|G)}[\log p(G|z)] - KL(q_\phi(z|G)||p(z))$. If we suppose $z = (z_j, z_g)$, then maximizing the ELBO can be formulated into maximizing the objective of VAE as in Eq. (4).

# B Proof of Theorem 1

*Proof.* We assume that the periodic graph is simulated from two latent factors as $G = \text{Sim}(F_l, F_g)$, where $F_l$ is the factor that is related to the local information, such as the structure of the repeating pattern and how repeating patterns are linked to each other. $F_g$ is defined as the factor of the global information, including how many repeating patterns the graph contains and their spatial arrangements. The goal is to prove that $z_l$ captures and only captures the information of $F_l$, and $z_g$ captures and only captures the information of $F_g$. Thus, based on the information theory, we need to prove $I(z_l, F_l) = H(F_l)$, $I(z_l, F_g) = 0$, $I(z_g, F_g) = H(F_g)$, and $I(z_g, F_l) = 0$. Here $I(a, b)$ refers to the mutual information between $a$ and $b$, and $H(*)$ refers to the information entropy of an element.

Based on the reconstruction error, after the model is well optimized, we can have all the latent variables to reconstruct the whole graph $G$. We also have $z_l \perp z_g$ and $F_l \perp F_g$ consider that the $z_l$ and $z_g$ are disentangled. Thus we have $I(z_l, F_l) + I(z_g, F_l) = H(F_l)$ and $I(z_l, F_g) + I(z_g, F_g) = H(F_g)$. Then we combine them to get

$$I(z_l, F_l) + I(z_l, F_g) + I(z_g, F_l) + I(z_g, F_g) = H(F_l) + H(F_g). \tag{1}$$

(1) First, we prove that $I(z_l, F_g) = 0$. Suppose we have two graphs $(G_1, G_2)$ with the same repeat pattern (i.e. $F_{l,1} = F_{l,2}$ and different global information (i.e. $F_{g,1} \neq F_{g,2}$). Regarding the latent variables, We have $I(z_{l,1}, F_{l,1}) + I(z_{l,1}, F_{g,1}) = H(z_{l,1})$ and $I(z_{l,2}, F_{l,2}) + I(z_{l,2}, F_{g,2}) = H(z_{l,2})$. Based on the condition (i.e. subject to) of the loss function, we enforce $z_{l,1} = z_{l,2}$. Thus, we have

$$I(z_{l,1}, F_{l,1}) + I(z_{l,1}, F_{g,1}) = I(z_{l,2}, F_{l,2}) + I(z_{l,2}, F_{g,2}). \tag{2}$$

Since $F_{l,1} = F_{l,2}$, we have $I(z_{l,1}, F_{l,1}) = I(z_{l,2}, F_{l,2})$. Then we should have $I(z_{l,1}, F_{g,1}) = I(z_{l,2}, F_{g,2})$. However, since $F_{g,1} \neq F_{g,2}$, the only situation to meet the requirement is $I(z_{l,1}, F_{g,1}) = I(z_{l,2}, F_{g,2}) = 0$. Thus, to generalize, for any graph, we have $I(z_l, F_g) = 0$.

(2) Second, we prove that $I(z_g, F_g) = H(F_g)$. Based on the conclusion that $I(z_l, F_g) = 0$ from last step, and the conclusion from the second paragraph that $I(z_l, F_g) + I(z_g, F_g) = H(F_g)$, we can get $I(z_g, F_g) = H(F_g)$.

(3) Third, we prove that $I(z_g, F_l) = 0$. We can rewrite the loss function as:

$$\max_{\theta,\phi} I(z_l, G) + I(z_g, G) \tag{3}$$

$$I(G, z_l) \leq I_1$$
$$I(G, z_g) \leq I_2$$

The detail of this derivation is provided in the work proposed by [2]. When $I_2$ is defined as $I_2 \leq H(F_g)$, we can have $I(G, z_g) \leq H(F_g)$. Since $G = \text{Sim}(F_l, F_g)$, it can be written as

$$I(z_g, F_l) + I(z_g, F_g) \leq H(F_g). \tag{4}$$

From the second step, we already have $I(z_g, F_g) = H(F_g)$. Thus, we have $I(z_g, F_l) = 0$.

(4) Forth, we prove $I(z_l, F_l) = H(F_l)$. Based on the conclusion from the first three steps, we have $I(z_l, F_g) = 0$, $I(z_g, F_g) = H(F_g)$, and $I(z_g, F_l) = 0$. When we combine these three three equations with Eq.(1), we can have $I(z_l, F_l) = H(F_l)$.

Given the above four steps, we have finally proved the four equations $I(z_l, F_l) = H(F_l)$, $I(z_l, F_g) = 0$, $I(z_g, F_g) = H(F_g)$, and $I(z_g, F_l) = 0$, which indicate that the latent vector $z_l$ capture and only capture local patterns and the latent vector $z_g$ capture and only capture global patterns. □

Table 1: Overview of datasets in experiments ($|\mathbf{G}|$ is the total number periodic graphs in the dataset; $|\mathcal{V}|_{avg}$ is the average graphs size of the dataset; $|\mathcal{E}|_{avg}$ is the average edges of the graph in the dataset; $|\mathbf{U}|$ is the types of basic units in the dataset; $|U|_{avg}$ is the average size of basic units in the dataset)

| Property | QMOF | MeshSeg | Synthetic |
|---|---|---|---|
| $|\mathbf{G}|$ | 3,780 | 300 | 46,500 |
| $|\mathcal{V}|_{avg}$ | 151.42 | 662.13 | 71.09 |
| $|\mathcal{E}|_{avg}$ | 1004.13 | 1747.24 | 107.34 |
| $|\mathbf{U}|$ | 14 | 1 | 3 |
| $|U|_{avg}$ | 18.93 | 3 | 4.33 |

Table 2: Implementation details of PGD-VAE. GIN represents the layer of Graph Isomorphism Network; ReLU represents the Rectified Linear Unit activation function; FC is the fully connected layer; Sigmoid is the sigmoid activation function.

| Layer | Local-pattern encoder | Global-pattern encoder | Local decoder | Neighborhood decoder | Global decoder | Assembler |
|---|---|---|---|---|---|---|
| Input | A | A | $z_l$ | $z_l$ | $z_g$ | $A^{(l)}, A^{(g)}, A^{(n)}$ |
| Layer1 | GIN+ReLU | GIN+ReLU | FC+ReLU | FC+ReLU | FC+ReLU | - |
| Layer2 | GIN+ReLU | GIN+ReLU | FC+ReLU | FC+ReLU | FC+ReLU | - |
| Layer3 | GIN+ReLU | GIN+ReLU | Sigmoid | Sigmoid | Sigmoid | - |
| Layer4 | Node clustering | Sum pooling | - | - | Replace diagonal with zero | - |
| Layer5 | Concatenate representative node embedding | - | - | - | - | - |
| Output | $z_l$ | $z_g$ | $A^{(l)}$ | $A^{(n)}$ | $A^{(g)}$ | A |

## C   Dataset

Two real-world datasets and one synthetic dataset were employed to evaluate the performance of PGD-VAE and other comparison models.

**QMOF dataset** The Quantum MOF (QMOF) is a publicaly available database of computed quantum-chemical properties and molecular structures of 21,059 experimentally synthesized metal–organic frameworks (MOF) [4]. In total 3,780 MOFs were selected for the experiment. The statistics of the QMOF dataset was summarized in Appendix, Table 1.

**MeshSeg dataset** The 3D Mesh Segmentation project contains 380 meshes for quantitative analysis of how people decompose objects into parts and for comparison of mesh segmentation algorithms [1]. Meshes in MeshSeg dataset can be formed into graphs of triangle grids. We made 10 replicates for each mesh. The statistics of MeshSeg dataset has been summarized in Appendix, Table 1.

**Synthetic dataset** The synthetic dataset contains three types of basic units: triangle, grid and hexagon. The statistics of the synthetic dataset has been summarized in Table 4. We augmented each basic unit to contain more basic units and finally we obtained 15,500 graphs for each local pattern. The statistics of synthetic dataset has been summarized in Appendix, Table 1.

## D   Implementation details of PGD-VAE

The implementation details of PGD-VAE are shown in Table 2. All comparison models were implemented by their default settings. The assembler has the form of matrix operation following the Eq. (3) in the main text and does not have a structure of neural network.

To solve the permutation invariance issue in graph generation, one can use BFS-based-ordering, which is commonly employed by existing works for graph generation such as GraphRNN [5] and GRAN [3]. The rooted node can be selected as the node with the largest node degree in the graph. Then BFS starts from the rooted node and visits neighbors in a node-degree-descending order. Such a BFS-based-ordering works well in terms of stability, which is similar to the situation in GraphRNN and GRAN.

## E   Case study details

In our case study, we distinguish atoms by their node degree within the graph of the unit cell (i.e., $A^{(l)}$). For instance, if the node has the degree of 2 in one unit cell, then it can be designed as an inorganic atom carrying a double positive charge (e.g., $Cu^{2+}$ or $Ag^{2+}$) connected to the surrounding organic atoms via metallic bonding, or a negative organic ion ($O^{2-}$) connecting with inorganic ions (e.g., $K^+$) or other organic atoms (e.g., $H^+$). Three MOFs have been illustrated in Figure 4 of the

main article, in which different atoms can be distinguished by the color and size. The figure is constructed via the ASE package by replacing the atom number with the degree of the nodes.

# F Evaluating Scalability

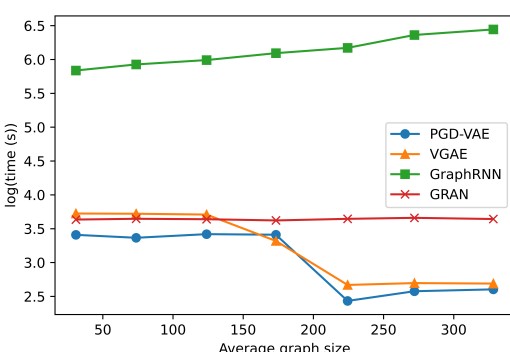

Figure 1: Scalability evaluation by running 10 epochs of PGD-VAE and comparison models.

We conducted experiments to evaluate the time complexity of PGD-VAE and comparison models, as shown in Figure 1 of Appendix. In the experiments, we recorded and logarithmized the time (s) to run 10 epochs on graphs in stratified synthetic dataset according to average graph size with PGD-VAE and comparison models. Aligned with the theoretical analysis, GraphVAE consumes the most of time compared with other models. Given the synthetic dataset, GRAN spends slightly more time than PGD-VAE and VGAE, while PGD-VAE and VGAE are less computational intensive and have comparable performance.

# G Generate atom types of QMOF data

We adapt PGD-VAE to be able to predict atom types of MOFs. Atom types are predicted by a prediction function modeled by MLP with the local embedding $z_l$ as the input. Since periodic graphs contain basic units as repeated patterns, we only need to predict atom types of a basic unit and assign them to other basic units. Two generated MOFs that contain two basic units from PGD-VAE are show in Figure 2.

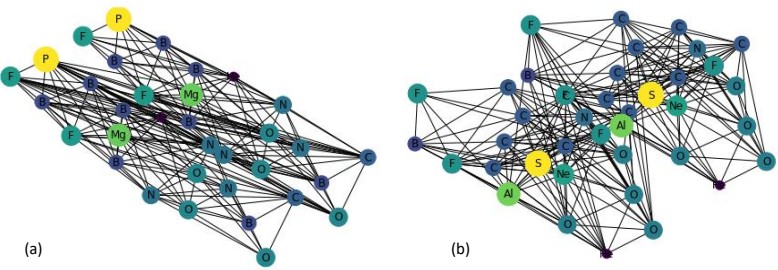

Figure 2: Generated MOFs from PGD-VAE. (a) $MgAl2B5F2O3C$; (b) $AlNeBSF3O4C7$