# OpenReview forum: "Deep Generative Model for Periodic Graphs"
_NeurIPS.cc/2022/Conference — NeurIPS 2022 Accept_

### Official Review · Reviewer_GS3z · 2022-07-06

**Rating:** 5
**Confidence:** 4
**Soundness:** 2 fair
**Presentation:** 2 fair
**Contribution:** 2 fair

**Summary:**

The paper proposes to tackle periodic graph generation, by learning to generate basic structural units and to compose them into a larger graph. This is achieved by using a hierarchical VAE approach with a loss term which encourages disentanglement.

**Questions:**

How exactly is m an n selected for each dataset? As I understood it, you use analytic methods (domain knowledge) to determine the optimal substructure sizes. This already seems to give a lot of ground-truth information for the model.

What ordering do the GraphRNN and GRAN use? The default BFS/DFS orderings that were originally used for these models or do you also provide them with the true graph ordering you use for your model?

In Figure 3, are the trees in the lower left corner part of the dataset? It is unclear which dataset this figure comes from. I presume it to be the synthetic one, but it does not have trees. If it is the synthetic dataset, it seems strange trees are generated even though all base units in the dataset where polygons.

**Limitations:**

As I mentioned in the weaknesses, the assumption that the adjacency matrix has a true canonical ordering is quite problematic. Even though for regular graphs in some cases we have something close to a canonical ordering. It essentially goes against all of the graph machine learning wisdom that is based on equivariance to the point that it is unclear we even need graph machine learning to solve this problem when this assumption trully holds.

**Strengths And Weaknesses:**

Mostly, current deep graph generation approaches aim to generate the graph by focusing on local structure so any attempt to introduce more focus on the global structure or decouple global and local structure generation is very welcome. Recently, a more general approach to decouple local and global structure generation, not focused on periodic graphs  has been proposed [1]. I think this should be mentioned in the related work. I haven't seen other work that focuses in particular on periodic graph generation using deep learning. Which makes this particular problem quite novel and potentially useful in problems such as crystal structure design.

What I find problematic, is the assumption that the node order in the graph is known and can be consistently determined.
First, it limits us to a small set of problems where we can even hope to determine such an ordering (e.g. atom lattices). Further, even for the problems considered here it's not entirely clear to me that we can determine such an ordering. For example lets consider the MeshSeq dataset. In this dataset we have a sequence of cuts people used to partition a given mesh. Even though the cuts are sequential, they gives us two equivalent parts (we build a binary tree, not a sequence) and depending on the particular mesh of the same type of object or its orientation which part is the "left" of the split and which is "right" could differ. Also, In general for mesh processing various equivariant architectures are normally used, due to the problem that in general we don't have a way to find a canonical orientation (ordering) of a mesh. So I don't see how we can get a canonical representation of a mesh here to use a loss that is not invariant to permutations.
In context of this an important reference [2] discusses, that for graph generation, you need a permutation invariant loss function. Which we do not have in the approach proposed here. I also think this reference should be included in the paper.
The second problem I see with assuming that we can get this canonical graph structure (ordering) is that in this case we are essentially just dealing with a set of matrices. So we can just use a traditional and very powerful matrix processing models such as convolutional neural networks. In fact the proposed architecture is a bit like a CNN already as it considers a bunch of patches and how they combine/overlap. So under this assumption I'm not even sure we need a GNN-based architecture.

A small note on the Global pattern encoder - it's true that using GIN and global pooling gives you 1-WL representation power, but 1-WL is inherently a local procedure, so I'm not sure that's the best architecture to capture the global graph structure. Models that usually perform better on graph-level tasks could be considered, such as the more expressive GNN architectures or graph transformers, which do not suffer from having only local interactions.

Why only two graph similarity measures (density and clustering coefficient) are used? Even the baselines that were used such as GRAN used more advanced measures, such as orbital counts or spectral features. Using only density and clustering coefficient are determined by local features, so there are no measures that show how well the global graph structure is reproduced. Which is one of the main goals of the work.

I'm also confused why the GraphVAE baseline was not used. It certainly has the closest overall architecture (GNN encoder + MLP decoder). Authors argue that it is too expensive, but the expense comes from the graph matching used to compute a permutation invariant loss. But authors claim, that for the graphs they consider we can assume the node order to fixed as otherwise their proposed method does not work. Also, GRAN used GraphVAE as a baseline on datasets with graphs as large as the ones considered in this paper (Protein graphs used in GRAN paper had up to 500 nodes). In GRAN paper they also just discarded GraphVAE matching loss and used the node ordering as given to GRAN model itself, which worked fine.

Writing quality could certainly be improved. The text is wordy, there's a lot of typos, sometimes words are missing in the sentences, even multiple words per sentence (e.g. line 125). I would strongly suggest the authors to at least use some sort of a spelling and grammar correction software (e.g. Grammarly).

[1] Martinkus et al. "SPECTRE: Spectral Conditioning Helps to Overcome the Expressivity Limits of One-shot Graph Generators." ICML 2022
[2] Vignac and Frossard "Top-N: Equivariant Set and Graph Generation without Exchangeability" ICLR 2022

---

> ### Author Response · Authors · 2022-08-06
> **Resposne to Reviewer 4 (GS3z) (Part I)**
>
> **Comments #1**: Recently, a more general approach to decouple local and global structure generation, not focused on periodic graphs has been proposed [1]. I think this should be mentioned in the related work. I haven't seen other work that focuses in particular on periodic graph generation using deep learning. Which makes this particular problem quite novel and potentially useful in problems such as crystal structure design.\
> **Response #1**:\
> Thank you for providing this information and we have surely added and discussed this paper in our related works section line 116-117.
>
> **Comments #2**: What I find problematic, is the assumption that the node order in the graph is known and can be consistently determined. First, it limits us to a small set of problems where we can even hope to determine such an ordering (e.g. atom lattices). Further, even for the problems considered here it's not entirely clear to me that we can determine such an ordering. ...... In context of this an important reference [2] discusses, that for graph generation, you need a permutation invariant loss function. Which we do not have in the approach proposed here. I also think this reference should be included in the paper. The second problem I see with assuming that we can get this canonical graph structure (ordering) is that in this case we are essentially just dealing with a set of matrices. So we can just use a traditional and very powerful matrix processing models such as convolutional neural networks. In fact the proposed architecture is a bit like a CNN already as it considers a bunch of patches and how they combine/overlap. So under this assumption I'm not even sure we need a GNN-based architecture.\
> **Response #2**:\
> (1) For now the proposed model borrows the same strategy as VGAE [1] to handle the order of nodes. Basically the graph is generated in a one-shot manner and without considering the graph matching. Suppose $A^{(l)}\in\mathbb{R}^{n\times n}$ is the adjacency matrix of basici unit, $A^{(g)}\in\mathbb{R}^{m\times m}$ is the adjacency matrix denoting the pairwise relations between basic units and $A^{(n)}\in\mathbb{R}^{n\times n}$ is the incidence matrix regarding how the nodes in adjacent basic units are connected. Even if we employ the graph matching algorithm in GraphVAE to solve the permutation invariance issue we can still achieve a better model complexity as we only need to match the order of nodes in $A^{(l)}$ and $A^{(n)}$. This corresponds to the time complexity of $O(m^4+n^4)$ which is much smaller than $O((mn)^4)$ beared by GraphVAE.\
> (2) GNN (e.g., GCN or GIN) is better to capture the local information of the graph since it captures the local information by aggregating the information of neighbors of each node in the graph. By contrast, CNN considers the adjacency matrix as a bunch of patches rather than a topological structure of the graph so that it basically just counts how many edges there are in a patch which might consist of many nodes without providing any topological information of the graph.\
> **Comment #3**: A small note on the Global pattern encoder - it's true that using GIN and global pooling gives you 1-WL representation power, but 1-WL is inherently a local procedure, so I'm not sure that's the best architecture to capture the global graph structure. Models that usually perform better on graph-level tasks could be considered, such as the more expressive GNN architectures or graph transformers, which do not suffer from having only local interactions.\
> **Response #3**:\
> (a) Thank you for the valuation comments. The proposed model is a general framework in that its encoder and decoder of the graph can be replaced with any alternatively depending on the needs. GIN, GCN and graph transformers are three options here as the encoder.
> (b) Although currently we use GIN as the encoder of our framework (PGD-VAE), we have achieved superior performance compared with other comparison models such as GraphRNN, GRAN and VGAE. As shown in Table 2, PGD-VAE achieves KLD smaller than comparison models by 0.8191 for clustering coefficient and 0.5062 for graph density on average on MeshSeq dataset, and 2.4275 for clustering coefficient as well as 1.7611 for graph density on average on Synthetic dataset. PGD-VAE also achieves better quality of graph generation regarding the uniqueness than GraphRNN and GRAN as shown in Table 3.\
>
> [1] Kipf, Thomas N., and Max Welling. "Variational graph auto-encoders." arXiv preprint arXiv:1611.07308 (2016).

---

> ### Author Response · Authors · 2022-08-06
> **Resposne to Reviewer 4 (GS3z) (Part II)**
>
> **Comment #4**: Why only two graph similarity measures (density and clustering coefficient) are used? Even the baselines that were used such as GRAN used more advanced measures, such as orbital counts or spectral features. Using only density and clustering coefficient are determined by local features, so there are no measures that show how well the global graph structure is reproduced. Which is one of the main goals of the work.\
> **Response #4**: We have added the metrics KLD of orbital between generated graphs and graphs in the training set in Table 2.\
> **Comment #5**: I'm also confused why the GraphVAE baseline was not used. It certainly has the closest overall architecture (GNN encoder + MLP decoder). Authors argue that it is too expensive, but the expense comes from the graph matching used to compute a permutation invariant loss. But authors claim, that for the graphs they consider we can assume the node order to fixed as otherwise their proposed method does not work. Also, GRAN used GraphVAE as a baseline on datasets with graphs as large as the ones considered in this paper (Protein graphs used in GRAN paper had up to 500 nodes). In GRAN paper they also just discarded GraphVAE matching loss and used the node ordering as given to GRAN model itself, which worked fine.\
> **Response #5**:\
> Thank you for the valuable suggestions of the review. We have added GraphVAE as the comparison model and the results are shown in Table 2 and Table 3.\
> **Comment #6**: Writing quality could certainly be improved. The text is wordy, there's a lot of typos, sometimes words are missing in the sentences, even multiple words per sentence (e.g. line 125). I would strongly suggest the authors to at least use some sort of a spelling and grammar correction software (e.g. Grammarly).\
> **Response #6**:\
> Thank you for the valuable comments, we have proofread the paper again and corrected all the typos in the paper.\
> **Comment #7**: How exactly is m an n selected for each dataset? As I understood it, you use analytic methods (domain knowledge) to determine the optimal substructure sizes. This already seems to give a lot of ground-truth information for the model.\
> **Response #7**:\
> The values of m (i.e., a safe estimate of largest number of basic units) and n (i.e., a safe estimate of largest number of nodes in each basic unit) are set as sufficiently-large values that are estimated following commonly-used manners in graph generation methods such as GraphVAE [1], VGAE [2], and GRAN [3]. In practice, the de facto estimation is readily achieved by the domain common sense. For example, to generate small molecules, GraphVAE estimates the largest number of nodes as 9 when applying it to QM9, a dataset of small molecules. GRAN estimates the largest number of nodes as 500 when applying it to Protein, a dataset of protein graphs where nodes are amino acids. Similarly, our m is set as 6 and n is set as 60 when being used in dataset MeshSeq, where there are less than dozens of number of nodes for each basic unit. Our m is set as 12 and n is set as 30 when being used in QMOF dataset which contains relatively larger basic units. We do NOT need analytic methods (domain knowledge) to determine the optimal substructure sizes. Moreover, the information of the number of nodes in each basic unit (and even the substructure) are commonly available in many periodic graph datasets and crystal structure datasets such as QMOF, OQMD, and Matbench. Hence it is even more convenient for us to use them when they are available.
>
> [1] Simonovsky, Martin, et al. "Graphvae: Towards generation of small graphs using variational autoencoders." International conference on artificial neural networks. Springer, Cham, 2018.\
> [2] Thomas N Kipf and Max Welling. 2016. Variational graph auto-encoders. NeurIPS’2016 Workshop (2016).
> [3] Liao, Renjie, et al. "Efficient graph generation with graph recurrent attention networks." Advances in neural information processing systems 32 (2019).

---

> ### Author Response · Authors · 2022-08-06
> **Resposne to Reviewer 4 (GS3z)**
>
> Thank you very much for your detailed summarization and insightful comments. Please find our answers to your comments/questions below. We have updated our paper based on your suggestions. If you have any further comments/suggestions on the updated version of our paper, we will be glad to improve on them. We also sincerely hope that the revised version and responses could help with an increase in the score.

---

> ### Author Response · Authors · 2022-08-06
> **Resposne to Reviewer 4 (GS3z) (Part III)**
>
> **Comment #8**: What ordering do the GraphRNN and GRAN use? The default BFS/DFS orderings that were originally used for these models or do you also provide them with the true graph ordering you use for your model?\
> **Response #8**:\
> In our work, we handle the permutation invariance by defining a canonical order of nodes. We can easily obtain the adjacency matrix ($A^{(l)}$) of a basic unit (i.e., unit cell for QMOF, two nodes linked by an edge for synthetic data and MeshSeg). Then these adjacency matrices of the single basic unit are put on the diagonal of the adjacency matrix A. The order of these basic units on the diagonal of A is determined by BFS. In addition, $A^{(n)}$ is fixed due to the periodicity of the graph. Once $A^{(l)}$, $A^{(n)}$ and the order of the basic units are fixed, then $A^{(g)}$ is fixed. A can be deterministically formed via $A^{(l)}$, $A^{(n)}$ and $A^{(g)}$. As a result, we can have a canonical order of periodic graph.
>
> **Comment #9**: In Figure 3, are the trees in the lower left corner part of the dataset? It is unclear which dataset this figure comes from. I presume it to be the synthetic one, but it does not have trees. If it is the synthetic dataset, it seems strange trees are generated even though all base units in the dataset where polygons.\
> **Response #9**:\
> (a) This figure comes from the model trained on the Synthetic dataset.
> (b) As shown in Figure 1, the basic units learned by our model are not adjacent with each other (i.e., share common nodes or edges). For instance, the basic unit for the synthetic data is just two nodes connected by an edge (line segment), indicating that the local pattern is well learned by our model. The branches shown in Figure 3 are just those basic units (line segments). The shape that they can form, such as triangle, square or polygon, is determined by $A^{(n)}$ learned by the neighborhood decoder.

---

> > ### Comment · Reviewer_GS3z · 2022-08-06
> > **Further Questions**
> >
> > Thank you for the explanation. How is the root node for the BFS selected? This presumably impacts the final ordering of the graph.

---

> ### Comment · Reviewer_GS3z · 2022-08-06
> **Response to Authors**
>
> Thank you for the detailed answers and improvements made to the paper. The lack of permutation equivariance still bugs me. That being said I have increased my score and if the BFS ordering procedure proves to be stable, I'll increase it one more point.
>
> Another note is that while the orbital counts are more informative than than clustering coefficient or degree counts it's still somewhat local (spectrum is better in this regard as it's truly global).

---

> > ### Author Response · Authors · 2022-08-07
> > **Further response to Comment #8**
> >
> > We sincerely thank the reviewer for giving us the opportunity to clarify this.
> >
> > To order the nodes in $A^{(l)}$ and $A^{(g)}$, we apply BFS-based-ordering, which is commonly employed by existing works for graph generation such as GraphRNN [1] and GRAN [2]. Here the rooted node is selected as the node with the largest node degree in the graph. Then BFS starts from the rooted node and visits neighbors in a node-degree-descending order. Such a BFS-based-ordering works well in terms of stability, which is similar to the situation in GraphRNN [1] and GRAN [2].
> >
> > (1) Suppose the graph has $n$ nodes in total. The number of possible node orderings of a graph is $n!$, but in practice BFS-based-ordering typically results in much smaller number of orderings, as explained in Section 2.3.4 in [1]. This means BFS typically leads to a much smaller number of node orderings than that led by random-ordering, which is what existing works observed in the experiments in real-world graphs [1].
> >
> > (2) We conducted additional experiments to demonstrate the stability of BFS-based orderings comparing to random-orderings that acts as a baseline. We randomly pick up 100 graphs from the QMOF dataset. For each graph $g_i$ (i=1,2,...,100), we randomly permute its nodes for 100 times and obtain 100 randomly permuted graphs $G$=\{$g_1, g_2, …, g_{100}$\}. (Hence in total we have 10,000 graphs). For each of these graphs, we apply the BFS as described above to order their nodes and obtain the corresponding BFS-ordered graphs $G’$=\{$g_1’, g_2’, …, g_{100}’$\}. Then we calculate Kendall rank correlation coefficient and Spearman's Rank Correlation coefficient along with relative p-values for each pair of graphs in G’. These two coefficients are commonly-used metrics for measuring similarity between orderings. Then, by comparison, for each graph in G we randomly order its nodes and obtain the corresponding randomly-ordered graphs $G’’$=\{$g_1’’, g_2’’, …, g_{100}’’$\}. We also calculate Kendall rank correlation coefficient and Spearman's Rank Correlation Coefficient along with relative p-values for each pair of graphs in G’’. The results regarding the average Kendall rank correlation coefficient, Spearman's Rank Correlation Coefficient, and p-values on both BFS-ordered graphs and randomly ordered graphs are shown below.
> >
> > | Data                          | Spearman's Rank Correlation |         | Kendall rank correlation |         |
> > |-------------------------------|-----------------------------|---------|--------------------------|---------|
> > |                               | Coefficient                 | p-value | Coefficient              | p-value |
> > | BFS-ordered graphs (G’)       | **0.6734**                      | **0.0114**  | **0.5742**                   | **0.0106**  |
> > | Randomly-ordered graphs (G’’) | 0.1243                      | 0.5040  | 0.0852                   | 0.5009  |
> >
> > Based on the results in the table, BFS-based-ordering achieves a Spearman's Rank correlation coefficient of 0.6734 which is much larger than the value of 0.1243 calculated from Randomly ordering. This indicates that although the input graphs come with very different node permutation, the BFS-based-ordering method can transfer them into very similar node orderings, which reveals the stability in BFS-based node ordering. Moreover, the low p-value of 0.0114 indicates that such stability is statistically significant. Similarly, BFS-based-ordering achieve a Kendall rank correlation coefficient of 0.5742 which is much larger than the value of 0.0852 calculated from Randomly ordering. This again indicates that although the input graphs come with very different node permutation, the BFS-based-ordering method can transfer them into very similar node orderings, which reveals the stability in BFS-based node ordering. Moreover, the low p-value of 0.0106 again indicates that such stability is statistically significant.
> >
> > We will add those results and discussions in our revised paper.
> >
> > [1] You, Jiaxuan, et al. "Graphrnn: Generating realistic graphs with deep auto-regressive models." International conference on machine learning. PMLR, 2018.
> >
> > [2] Thomas N Kipf and Max Welling. 2016. Variational graph auto-encoders. NeurIPS’2016 Workshop (2016). [3] Liao, Renjie, et al. "Efficient graph generation with graph recurrent attention networks." Advances in neural information processing systems 32 (2019).

---

> > > ### Comment · Reviewer_GS3z · 2022-08-07
> > > **Response to Authors**
> > >
> > > Thank you. They seem not perfectly stable, but as you say not terrible. As promised I increased my score.

---

### Official Review · Reviewer_9Uud · 2022-07-08

**Rating:** 5
**Confidence:** 3
**Soundness:** 3 good
**Presentation:** 2 fair
**Contribution:** 2 fair

**Summary:**

This paper focuses on the problem of periodic graph generation, and proposes PGD-VAE, a Variational AutoEncoder based generative model for periodic graphs that can automatically learn, disentangle, and generate local and global patterns. The authors formulate the generation of periodic graphs as the generation of a finite number of basic units.

**Questions:**

* If the goal is to make the generation of local structures and global repeating patterns independent, why do you  introduce a contrastive loss to make the generation of local structures conditioned on the global repeating patterns?

**Limitations:**

Yes, they addressed the limitations of the proposed method.

**Strengths And Weaknesses:**

## Strengths
* originality: This paper formulates a new problem setting for periodic graph generation. Instead of generating basic units and repeating patterns, they generate the periodic graph consist of a fixed number of basic units.

## Weaknesses
* According to the authors, this paper's main idea of the generation process is to make the generation of local structures and global repeating patterns independent. But they introduce a contrastive loss to make the generation of local structures actually **conditioned** on the global repeating patterns.
* In this paper, the generation of node type such as atom type in QMOF is ignored. Only the structure is generated.
* Permutation invariance is ignored. The proposed method can not satisfy permutation invariance when the node type needs to be considered. The application of the proposed method is limited because of breaking permutation invariance.
* Because of above considerations, the quality and significance of the proposed method is limited.
* Clarity: Hard to read. There are variables coming from nowhere and not shown in the equation, such as Equation (8). Many sentences hard to understand.

---

> ### Author Response · Authors · 2022-08-02
> **Response to Reviewer 3 (9Uud)**
>
> **Comment #1**: According to the authors, this paper's main idea of the generation process is to make the generation of local structures and global repeating patterns independent. But they introduce a contrastive loss to make the generation of local structures actually conditioned on the global repeating patterns.\
> **Response #1**:\
> (1) We don’t have the term “global repeating patterns” in our paper. We only have $z_g$ that captures global information of the graph and $A^{(g)}$ that denotes the pairwise relations among basic units. None of them are related to the calculation of the contrastive loss.\
> (2) The local structures are not conditioned on the global one. The contrastive loss is computed by only enforcing the representation of local structures ($z_l$) equal for graphs with the same basic unit and different for those with different basic units; as shown in the Equation of contrastive loss below, it has nothing to do with the global $z_g$.\
> **Comment #2**: In this paper, the generation of node type such as atom type in QMOF is ignored. Only the structure is generated.\
> **Response #2**: \
> Although generating node attribute is not the focus of our paper, we are still working on to add experiments to predict atom types of generated MOF simultaneously when generating the graph topology and will update our results shortly.\
> **Comment #3**: Permutation invariance is ignored. The proposed method can not satisfy permutation invariance when the node type needs to be considered. The application of the proposed method is limited because of breaking permutation invariance.\
> **Response #3**\
> For now the proposed model borrows the same strategy as VGAE [1] to handle the order of nodes. Basically the graph is generated in a one-manner and without considering the graph matching. Suppose $A^{(l)}\in\mathbb{R}^{n\times n}$ is the adjacency matrix of basici unit, $A^{(g)}\in\mathbb{R}^{m\times m}$ is the adjacency matrix denoting the pairwise relations between basic units and $A^{(n)}\in\mathbb{R}^{n\times n}$ is the incidence matrix regarding how the nodes in adjacent basic units are connected. Even if we employ the graph matching algorithm in GraphVAE to solve the permutation invariance issue we can still achieve a better model complexity as we only need to match the order of nodes in $A^{(l)}$ and $A^{(n)}$. This corresponds to the time complexity of $O(m^4+n^4)$ which is much smaller than $O((mn)^4)$ beared by GraphVAE.\
> **Comment #4**: Clarity: Hard to read. There are variables coming from nowhere and not shown in the equation, such as Equation (8). Many sentences hard to understand.\
> **Response #4**: \
> The notations in Eq. (8) have been explained in Section 3.11, line 197 and line 203-204.\
> **Comment #5**: If the goal is to make the generation of local structures and global repeating patterns independent, why do you introduce a contrastive loss to make the generation of local structures conditioned on the global repeating patterns?\
> **Response #5**:\
> (1) We don’t have the term “global repeating patterns” in our paper. We only have $z_g$ that captures global information of the graph and $A^{(g)}$ that denotes the pairwise relations among basic units. None of them are related to the calculation of the contrastive loss.\
> (2) The local structures are not conditioned on the global one. The contrastive loss is computed by only enforcing the representation of local structures ($z_l$) equal for graphs with the same basic unit and different for those with different basic units; as shown in the Equation of contrastive loss below, it has nothing to do with the global $z_g$.\
>
> [1] Kipf, Thomas N., and Max Welling. "Variational graph auto-encoders." arXiv preprint arXiv:1611.07308 (2016).

---

> > ### Comment · Reviewer_9Uud · 2022-08-05
> > **Concern about the contrastive loss is addressed**
> >
> > My concern about the contrastive loss is addressed. Therefore, I will increase my score.

---

> > > ### Author Response · Authors · 2022-08-08
> > > **Further response to Comment #2**
> > >
> > > Thank you for your precious time and suggestions for us to improve our work!
> > >
> > > We conduct additional experiments to show the application of our model by adding a prediction function (modeled by MLP) to predict atom types of MOFs using QMOF dataset. Note that periodic graphs contain repeated basic units so that we only need to predict atom types for one basic unit and assign them to other basic units. The results have been added to Appendix Section G.

---

### Official Review · Reviewer_WzHK · 2022-07-11

**Rating:** 8
**Confidence:** 5
**Soundness:** 4 excellent
**Presentation:** 3 good
**Contribution:** 4 excellent

**Summary:**

Generating novel periodic graphs is an interesting and important task as the application examples given in the paper. Three challenges towards this task are provided by the authors, which seem reasonable to me. To address these challenges, the authors propose a novel model to generate periodic graphs by automatically learning, disentangling, generating and assembling local and global patterns of periodic graphs. This idea of “hierarchically”generating periodic graphs is novel and it is suitable and efficient to handle this type of graphs as suggested by the complexity analysis part. In addition, the experimental results show the effectiveness of this model compared with other comparison models.


**Questions:**

1. The author can add and discuss the missing references, such as [1] in the Related Works session.
2. Please check and fix the typos, e.g., that in Figure 2.


**Limitations:**

The limitations of the proposed work are discussed in the Conclusion session.
This work does not have potential negative societal impacts.

**Strengths And Weaknesses:**

Strengths:
1 The framework proposed by the paper solves an important task to generate periodic graphs and is novel to me as it generates periodic graphs in a hierarchical fashion. This strategy looks natural since the periodic graph (e.g., graph structure of materials) can be very huge and is intuitively more efficient since it only needs to learn and generate local and global patterns. Also, to my understanding, the node embedding clustering module embedded in the local-pattern encoder can learn the local structure of the graph and perform at least as well as the sum pooling on the whole graph.
2 The analysis on the model is comprehensive both theoretically and experimentally. The learning objective has been derived properly. The complexity of the model is better than the comparison models as shown in Table 1. The proposed model also has better performance than the comparison models.
3 The presentation of the paper is good to me. Figure 2 is organized very well and is intuitive to show how the framework functions. The ability to disentangle local and global patterns has been demonstrated by Figure 3.
4 The authors published the code.


Weakness:
1 Some typos: for example, in Figure 2, it should be A^{(l)}, A^{(g)} and A^{(n)} rather than A_l, A_g and A_n.
2 Missing references: for example, [1] below aims to generate periodic structures, although seems using different strategies and different type of input data.

[1] Xie, Tian, et al. "Crystal diffusion variational autoencoder for periodic material generation." arXiv preprint arXiv:2110.06197 (2021).

---

> ### Author Response · Authors · 2022-08-06
> **Response to Reviewer 2 (WzHK)**
>
> Thank you very much for your detailed summarization and insightful comments. Please find our answers to your comments/questions below. We have updated our paper based on your suggestions. If you have any further comments/suggestions on the updated version of our paper, we will be glad to improve on them. We also sincerely hope that the revised version and responses could help with an increase in the score.
>
> **Comment #1**: Some typos: for example, in Figure 2, it should be A^{(l)}, A^{(g)} and A^{(n)} rather than A_l, A_g and A_n.\
> **Response #1**:\
> Thank you for pointing this out. We have addressed those typos in the revised paper.
>
> **Comment #2**: Missing references: for example, [1] below aims to generate periodic structures, although seems using different strategies and different type of input data.\
> **Response #2**:\
> Thank you for guiding us to this paper. Indeed this one focus on the generation of periodic graphs and we will surely add it to our paper.

---

> > ### Comment · Reviewer_WzHK · 2022-08-09
> > **Responses**
> >
> > My comments have been properly addressed.

---

### Official Review · Reviewer_bnmq · 2022-07-13

**Rating:** 6
**Confidence:** 3
**Soundness:** 3 good
**Presentation:** 3 good
**Contribution:** 3 good

**Summary:**

This work proposed a novel VAE-based method to generate periodic graphs with applications to MOF, mesh and synthetic graphs generation. The proposed VAE-based method (PGD-VAE) disentangles the graph representation into local and global semantics by generating $A^{(l)}, A^{(g)}, A^{(n)} $ and then assembles them into a complete periodic graph. Overall, I enjoyed reading the paper. But there are some concerns to be addressed.

**Questions:**

Please refer to the *weaknesses* for my questions.

**Limitations:**

The limitation of this work is that the generated periodic graphs may not be applicable to real-world applications.

**Strengths And Weaknesses:**

Strengths:
* The paper is well-written and easy to follow
* To the best of my knowledge, the proposed PGD-VAE is novel in generating periodic graphs, and the proposed disentangled periodic graph representation is interesting.
* The results are impressive on the evaluation metrics.

Weaknesses:
* Why not use the disentangled periodic graph representation as input to the PGD-VAE as well?
* More Ablation studies on the necessity of disentanglement should be added to validate the claim of the benefits of using disentangled generation.
* The generation results look good on the evaluation metrics. However, it is unknown how the generated graphs can be applicable to real-world applications. For instance, are the generated MOF structures stable and synthesizable?

---

> ### Author Response · Authors · 2022-08-02
> **Response to Reviewer 1 (bnmq)**
>
> **Comment #1**: Why not use the disentangled periodic graph representation as input to the PGD-VAE as well? \
> **Response #1**:\
> The inputs of PGD-VAE, namely,  $z_l$ and $z_g$, are disentangled periodic graph representations. Specifically, $z_l$ and $z_g$ are disentangled from each other. and the variables in $z_l$, which correspond to different basic units, are also disentangled from each other.
>
> **Comment #2**: More Ablation studies on the necessity of disentanglement should be added to validate the claim of the benefits of using disentangled generation. \
> **Response #2**:\
> (1) We have conducted the ablation study by removing the regularization term (contrastive loss) which is responsible to achieve the disentanglement of $z_l$ corresponding to different unit cells and the disentanglement between $z_l$ and $z_g$, as shown in Table 2.\
> (2) We also have added the discussions in the paper:\
> “The results in Table 2 shows that without disentanglement the model (PGD-VAE-1) achieves 0.4630 and 0.4921 of KLD on clustering coefficient and density for MeshSeq dataset, respectively, which is worse than those obtained from the full model (PGD-VAE) by 0.0653 and 0.0386. In addition, PGD-VAE-1 achieves 3.2672 and 0.8399 of KLD on clustering coefficient and density for Synthetic dataset, which is worse than the full model by 0.0809 and 0.1575 as well.”
>
> **Comment #3**: The generation results look good on the evaluation metrics. However, it is unknown how the generated graphs can be applicable to real-world applications. For instance, are the generated MOF structures stable and synthesizable?\
> **Response #3**:\
> (1) Generation of periodic graphs can be applied to a few real-world domains since periodic graphs naturally characterize many real-world applications ranging from crystal nets containing repetitive unit cells to polygon mesh data containing repetitive grids. We have discussed applications in line 29-31 and presented its forms in the real world Figure 1.\
> (2) Although generating node attribute is not the focus of our paper, we are still working on to add experiments to predict atom types of generated MOF simultaneously when generating the graph topology and will update our results shortly.

---

> ### Author Response · Authors · 2022-08-08
> **Further response to Comment #3**
>
> Thank you for your precious time and patience!
>
> We conduct additional experiments to show one potential application of our model by adding a prediction function (modeled by MLP) to predict atom types of MOFs using QMOF dataset. Note that periodic graphs contain repeated basic units so that we only need to predict atom types for one basic unit and assign them to other basic units. The results have been added to Appendix Section G.

---

### Author Response · Authors · 2022-08-09
**A final summary of updates in the paper**

We sincerely appreciate for reviewers' comments and feedbacks that made our paper further improved when we were addressing their concerns. We are also glad that reviewers approved our clarifications and are satisfied with how we addressed their comments. The following provides a final summary of the updates:

1. We have addressed all typos in the revised paper.
2. We have discussed the paper "Crystal diffusion variational autoencoder for periodic material generation" in line 129-130 as suggested by reviewer 2.
3. We have discussed the paper "SPECTRE: Spectral Conditioning Helps to Overcome the Expressivity Limits of One-shot Graph Generators." in line 116-117 as suggested by reviewer 4.
4. We have showcased the application of our model by predicting atom types using QMOF dataset and presented the results in Appendix Section G.
5. We have conducted additional experiments by adding GraphVAE as the comparison model and the results are shown in Table 2 and Table 3.
6. We have conduced additional experiments to show the stability of BFS ordering of nodes and relavent discussions in Appendix Section H.

---

### Meta-Review · Area_Chair_UzbP · 2022-08-25

**Recommendation:** Accept
**Confidence:** Certain

**Metareview:**

This paper proposed an interesting model for generating periodic graphs.  The hierarchical representation of periodic graphs decomposes into local structure and global structure, and greatly reduces the size of the structure to be modeled, which is an interesting contribution.  All reviewers liked the idea of this representation and lean toward acceptance.

I would, however, encourage the authors to include some discussion of the limitations of this approach, in particular the dependence on a stable node ordering, and the fact that the model can only generate perfectly periodic graphs (what if the graph is mostly periodic but with some noise?).

**Award:**

No

---

### Decision · Program_Chairs · 2022-09-14

Accept